

# Simulating the spread of disinfection by-products and anthropogenic bromoform emissions from ballast water discharge in Southeast Asia

Josefine Maas[1], Susann Tegtmeier[1], Birgit Quack[1], Arne Biastoch[1,2], Jonathan V. Durgadoo[1], Siren Rühs[1], Stephan Gollasch[3], Matej David[4,5]

[1]GEOMAR Helmholtz Centre for Ocean Research Kiel, Kiel, Germany
[2]Christian-Albrechts-Universität zu Kiel, Kiel, Germany
[3]GoConsult, Hamburg, Germany
[4]Dr. Matej David Consult, Izola, Slovenia
[5]Faculty of Maritime Studies, University of Rijeka, Croatia

*Correspondence to*: Josefine Maas (jmaas@geomar.de)

**Abstract.** Ballast water treatment is required for vessels to prevent the introduction of potentially invasive neobiota. Some treatment methods use chemical disinfectants which produce a variety of halogenated compounds as disinfection by-products (DBPs). One of the most abundant DBP from oxidative ballast water treatment is bromoform ($CHBr_3$) where we find an average concentration of $894 \pm 560$ nmol $L^{-1}$ ($226 \pm 142$ µg $L^{-1}$) in the undiluted ballast water from measurements and literature. Bromoform is a relevant gas for atmospheric chemistry and ozone depletion, especially in the tropics where entrainment into the stratosphere is possible. The spread of DBPs in the tropics over months to years is assessed here for the first time. With Lagrangian trajectories based on the NEMO-ORCA12 model velocity field, we simulate DBP spread in the sea surface and try to quantify the oceanic bromoform concentration and emission to the atmosphere from ballast water discharge at major harbours in the tropical region of Southeast Asia. The exemplary simulations of two important regions, Singapore and the Pearl River Delta, reveal major transport pathways of the DBPs and the anthropogenic bromoform concentrations in the sea surface. Based on our simulations, we expect DBPs to spread into the open ocean, along the coast and also an advection with monsoon-driven currents into the North Pacific and Indian Ocean. Furthermore, anthropogenic bromoform concentrations and emissions are predicted to increase locally around large harbours. In the sea surface around Singapore we estimate an increase in bromoform concentration by 9 % compared to recent measurement. In a moderate scenario where 70 % of the ballast water is chemically treated bromoform emissions to the atmosphere can locally exceed 1000 pmol $m^{-2}$ $h^{-1}$ and double climatological emissions. In the Pearl River Delta all bromoform is directly outgassed which leads to an additional bromine (Br) input into the atmosphere of 495 kmol Br (~42 t $CHBr_3$) $a^{-1}$. From Singapore ports the additional atmospheric Br input is calculated as 312 kmol Br (~26 t $CHBr_3$) $a^{-1}$. We estimate the global anthropogenic Br input from ballast water into the atmosphere of up to 13 Mmol $a^{-1}$. This is 0.1 % global Br input from background bromoform emissions and thus probably not relevant for stratospheric ozone depletion.



# 1 Introduction

## 1.1 Ballast water treatment

Ballast water is necessary for ships to maintain stability and draught during voyage and port operations. Usually ballast water is taken up during cargo unloading and discharged during loading operations. However, the uptake and discharge of ballast water by commercial ships is the main driver of the global spread of marine invasive species, which can cause negative impacts on the ecosystems, economies, and public health (Ruiz et al., 2000; Briski et al., 2012). In September 2017, the Ballast Water Management Convention (IMO, 2004) entered into force aiming to minimise the survival of organisms carried in ballast water tanks. According to the convention, shipowners from ratified flag states have different options to manage their ballast water, one being the on-board operation of a ballast water treatment system (BWTS) that is type approved by the member states. Over the next years, more than 75,000 vessels have to install such BWTS in order to control the transport of potentially harmful species (David and Gollasch, 2015).

Different BWTS are available which can be separated into physical and chemical oxidation methods (e.g. David and Gollasch, 2015). Physical methods include filtration, cavitation and treatment with ultra-violet radiation. Chemical treatment is achieved via, e.g. electrolysis, chlorination or ozonation. Electrolysis makes use of electricity in the ballast water to generate sodium hypochlorite as oxidant from the chemical reaction of salt in the seawater. During chlorination, a chemical such as sodium hypochlorite or chlorine dioxide is added in solution to the ballast water. Until 2017, 75 systems received approval of which 33 use chemical treatment. Chlorination techniques, such as electrolysis and chlorination, are expected to be the dominating treatment method with 20 systems using some form of chlorination. However, the exact share of the different treatment methods that will be installed on ships over the next years is currently unknown.

Chemical BWTS that apply oxidative treatment have been shown to produce a variety of so-called disinfection by-products (DBPs) including trihalomethanes, halogenated acetic acids and bromate (Delacroix et al., 2013; Werschkun et al., 2014; Shah et al., 2015). The generally proposed mechanism for generating DBPs is the reaction of oxidants such as chlorine and ozone, with organic and inorganic substances, such as bromide (Br-) and iodide (I-), in the water via the formation of hypobromous (HOBr), and hypoiodous (HOI) acid. The nature and amount of DBPs generated in seawater depend on many factors including the type of oxidant, the injected concentration, the amount and composition of dissolved organic matter (DOM) (Liu et al., 2015) and the concentrations of the specific halide ions. Chlorination of seawater, for example, has been shown to produce bromoform ($CHBr_3$) as one major DBP (Jenner et al., 1997; Padhi et al., 2012; Liu et al., 2015). It will spread with many other DBPs in the marine environment once the ship discharges its ballast water. A recent risk assessment with the methodology of the Joint Group of Experts on the Scientific Aspects of Marine Environmental Protection – Ballast Water Working Group (GESAMP-BWWG) showed, that this methodology does not fully account for the potential environmental risks (David et al., 2018).



## 1.2 Brominated very short-lived substances

Trihalomethanes generated in BWTS such as bromoform are also formed naturally in the oceans. Bromoform is of biological
origin with both macro- and microalgae as potential producers, which oxidise primary metabolic compounds with
haloperoxidases in the presence of hydrogen peroxide (e.g. Theiler et al., 1978; Moore et al., 1996). Currently available
measurements of bromoform in seawater suggest a large spatial variability with elevated abundances in coastal, equatorial
and upwelling regions due to biological sources (Quack and Wallace, 2003; Ziska et al., 2013; Fuhlbrügge et al., 2016).
Bromoform is the most important carrier of organic bromine from the ocean to the atmosphere, contributing together with
dibromomethane ($CH_2Br_2$) up to 70 % of organic bromine to the marine troposphere (Hossaini et al., 2012). Both compounds
have relatively short lifetimes of around two weeks ($CHBr_3$) and three months ($CH_2Br_2$) in the tropical boundary layer of the
atmosphere and thus belong to the so-called very short-lived substances (VSLS) (Carpenter and Reimann et al., 2014). Given
the highly variable oceanic production and its short lifetime, the atmospheric distribution of bromoform is characterised by
strong variations (Quack and Wallace, 2003).

Upon their release into the atmosphere, bromoform and other brominated VSLS impact atmospheric chemistry. VSLS are
quickly oxidised or photodissociated to reactive halogen species, which participate in the depletion of tropospheric ozone by
catalytic cycles (Saiz-Lopez and von Glasow, 2012). Furthermore, reactive halogen species alter tropospheric $HO_x$ and $NO_x$
ratios by acting directly as oxidants (e.g., for dimethylsulfide (DMS) and mercury). Thereby oceanic VSLS impact the
atmospheric lifetimes of many pollutants and greenhouse gases, most importantly methane.

In the stratosphere, VSLS also contribute to the depletion of ozone. Due to their short lifetime, they are mostly oxidised and
subsequently removed through tropospheric precipitation. However, in regions of deep convection, they can be entrained into
the stratosphere through rapid vertical transport (e.g., Aschmann et al., 2009; Tegtmeier et al., 2015). Deep convective events
are most common in the tropics near the equator where solar irradiance is high throughout the year and the ocean is an
efficient source of bromoform and other VSLS (Quack and Wallace, 2003). Observational (e.g., Dorf et al., 2006) and
modelling (e.g., Warwick et al. 2006; Liang et al., 2010;) studies have suggested that VSLS provide a significant contribution
to stratospheric total bromine ($Br_y$) with current estimates ranging between 2 to 8 ppt (Carpenter and Reimann et al., 2014;
Wales et al., 2018). Once brominated VSLS have reached the stratosphere, they participate in ozone depletion at middle and
high latitudes (Yang et al., 2014; Sinnhuber and Meul, 2015).

## 1.3 Motivation

Recent publications have analysed the production of DBPs from oxidative ballast water treatment and assessed its
ecotoxicity (Delacroix et al., 2013; Shah et al., 2015; Werschkun et al., 2014). These studies focussed on the risk assessment
on board of ships or the near-ship environment. Until now, the focus was on the small-scale immediate exposure of DBPs to
humans and marine environment. The long-term effects of ballast water discharge on regional to global scales has not been





assessed so far and the atmosphere, as a sink for volatile halocarbons, has not been considered in any existing risk

assessment of oxidative ballast water treatment. In particular, brominated species such as bromoform are frequently produced in treated ballast water and are known to impact atmospheric chemistry. In this study, we provide a first analysis of how DBPs or any other passive substance contained in treated ballast water spreads over a period of months to years around different harbours in Southeast Asia (Section 3). The derived spread can serve as a proxy for assessing the environmental impact of any chemical or biological species contained in treated or untreated ballast water. In a second step, we will derive

an estimate of bromoform released from ballast water into the marine environment and quantify its emission into the atmosphere (Section 4). We will further discuss the methods and data used for this study including port statistics, DBP concentrations in ballast water and our Lagrangian simulations (Section 2).

## 2 Methods

### 2.1 Port statistics

For a regional to global analysis, volumes of ballast water discharge for individual ports and the typical bromoform concentration in the treated ballast water are required. As there are no exact numbers of ballast water discharge volume available for most ports, we derive estimates of the discharge volumes by linking the annual amount of global ballast water volume with the cargo throughput at each port. Global annual discharge volume is estimated to range between 3 to 5 billion m³ (Tamelander et al., 2010; Endresen et al., 2004; David, 2015). In addition to the global ballast water amount, it is

known that the discharged ballast water of each ship amounts on average to roughly 33 % of its cargo volume (David, 2015). Here, we use the global ballast water discharge as 33 % of the global 10286.9 million tons of loaded goods (UNCTAD, 2017) to obtain a ballast water volume of 3.4 billion $m^3$ which agrees well with the estimates from the studies mentioned above.

The cargo throughput is obtained from the world port ranking 2016 published by the American Association of Port

Authorities (AAPA, 2016). This statistic includes the hundred biggest ports for two categories, containerised and bulk cargo, respectively. Containerised cargo is given in twenty-foot equivalent units (TEUs), while bulk cargo is given in tons. In order to combine these two rankings, we generate a modified world port ranking and calculate the percentage share of containerised and tonnage goods according to their global ratio given by the Review of Maritime Transport (UNCTAD, 2017). The percentage share of containerised goods in 2016 amounts to 16.6 %, while the rest makes up 83.4 % of the total

goods loaded (UNCTAD, 2017). According to the percentage share of their category and their individual size in the ranking, each container and bulk port is assigned its relative fraction, under the simplified assumption that these ports account for all of the global commercial ship trade and receive all of the global ballast water. Since many of the harbours appear in both statistics, containerised and bulk, both percentage values are added to give their total cargo share, which forms our modified





world port ranking encompassing 144 ports (Supplement Table). We use the calculated percentages to divide global ballast
water volume among all ports to derive the estimated discharge for each port.

Our study focus was set on the coastal region of Southeast Asia where 38 harbours from our modified world port ranking are
located comprising 57 % of the global shipping industry (Figure 1). This region is to a major extent located in the tropics
which makes it very relevant for entrainment of oceanic VSLS emissions into the stratosphere.
For our detailed analysis, we choose two different locations in the tropics that are characterised by large harbours and
different ocean dynamics. The first area is Singapore, where the two ports of Singapore and Tanjung Pelepas (Malaysia) are
located very close to the equator. This location in the Maritime continent is characterised by sea surface currents of over
$0.2 \text{ m s}^{-1}$ in the climatological mean (Figure 1). The other location is the Pearl River Delta (PRD) where the harbour cities of
Guangzhou, Hong Kong and Shenzhen are located. There, only weak coastal currents can be found in the climatology.

**2.2 Bromoform production from oxidative ballast water treatment**

In a second step, we derive estimates of the bromoform concentration produced during chemical ballast water treatment.
Since there are many BWTS that use different chlorination techniques with different water parameters and residence times,
the produced amount of bromoform can show large variations. Here, we determine a range of possible bromoform
concentrations which can be used in our analysis to estimate the environmental input of bromoform. For this purpose we use
measurements of chemically treated ballast water taken during shipboard tests, as well as literature data. The formation of
disinfection by-products in BWTS is most commonly investigated during land-based tests. In contrast, we have conducted
one of the first shipboard tests of the formation of major halocarbons in treated ballast water. The samples were taken from
the discharge of treated ballast water of three unnamed BWTS, two in Norway and one in Germany, which use chlorination
techniques. This allows us to obtain a more robust estimate of the initial bromoform concentrations in the ballast water. The
bromoform measurements were carried out with a Purge-and-Trap chromatograph/mass spectrometer (GC/SM) system with
a detection limit of around $0.1 \text{ pmol L}^{-1}$. Bromoform concentrations of $244.5 \pm 163.6 \text{ µg L}^{-1}$ ($967.6 \pm 647.4 \text{ nmol L}^{-1}$) were
found in 12 ballast water samples taken in Norway from two different BWTS (Table 1). Bromoform concentrations of
$202.0 \pm 74.0 \text{ µg L}^{-1}$ ($799.1 \pm 292.7 \text{ nmol L}^{-1}$) were found in 9 ballast water samples taken in Germany from one BWTS
(Table 1). These samples were taken on board the vessel at three time periods during ballast water discharge in 15-20
minutes intervals. The particulate organic matter in the water was 11.1 to $12.6 \text{ mg L}^{-1}$.
In addition, we use bromoform concentrations given in the reports of the International Maritime Organization (IMO) Marine
Environment Protection Committee (MEPC) for Final Approval of BWTS (docs.imo.org). Mean values from the MEPC
reports corresponding to 29 BWTS for seawater and brackish water are also given in Table 1.



Different chemical treatment systems show greatly varying bromoform concentrations as illustrated by the large standard deviations in the MEPC data. This is due to different doses of oxidant, varying residence time in the tank as well as different water properties such as salinity, temperature and the amount of DOM (Shah et al., 2015). Samples of the same treatment system (German system in Table 1) show a smaller standard deviation. Overall, our shipboard DBP measurements are in the range of the land-based test results published in the MEPC reports, suggesting a similar amount of bromoform production.

On average $226 \pm 142$ µg $CHBr_3$ $L^{-1}$ can be expected in ballast water, which corresponds to $894 \pm 560$ nmol $L^{-1}$ (Table 1) with the mean values of all four data sets in good agreement.

The exact percentage of vessels that will eventually use chlorination BWTS is unknown. Oxidative water treatment is more suited for larger vessels such as bulk carriers or tankers, typically those types of ships that carry the largest volumes of

ballast water (Maritime Impact, 2017). Thus, we assume that $70 \pm 20$ % of the ballast water will be chemically treated producing DBPs. In order to capture the range of uncertainty resulting from the variations of the bromoform concentrations in ballast water samples and from the unknown share of chemical BWTS, we set up three scenarios: LOW, MODERATE and HIGH (Table 2). The scenarios are assigned an initial bromoform concentration corresponding to the mean and the mean $\pm$ one standard deviation, respectively, and they use different shares (50, 70 and 90 %) of chemically treated ballast water.

Based on these two assumptions, we derive different amounts of annually discharged bromoform for the selected regions at Singapore and Pearl River Delta (Table 2).

**2.3 Lagrangian simulations**

In contrast to earlier studies which focussed on the local effect of DBPs from ballast water (e.g. David et al. 2018), we investigate the long-term, large-scale influence of DBPs in the ocean and atmosphere. Therefore, we need regional to global

ocean velocity and surface wind data which can be obtained from high-resolution ocean general circulation models (OGCM). We simulate the spread of treated ballast water and the DBPs contained within, by applying a Lagrangian trajectory integration scheme to the 3D velocity output from an eddy-resolving OGCM. The model output stems from a hindcast experiment with the model configuration based on the NEMO-ORCA code version 3.6 (Madec, 2008). The ORCA0083 configuration from the European DRAKKAR consortium (The DRAKKAR Group, 2007) has a horizontal

resolution of 1/12° degrees and 75 vertical levels, with 46 levels in the upper 1000 m and spacing increasing with depth (see also Marzocchi et al., 2015; Durgadoo et al., 2017). Atmospheric forcing comes from the DFS5.2 data set (Dussin, 2016) and varies on a range of scales, from synoptic to interannual and longer. The experiment ORCA0083-N06 used in this study was run by the National Oceanography Centre, Southampton, UK. Model output is given at a temporal resolution of five days for the time period 1963 to 2012.



For simulating the spread of DBPs in the surface ocean, the ARIANE software was used (Blanke et al., 1999). ARIANE performs offline trajectory calculations by passively advecting virtual particles along analytically computed 3D streamlines. This method has been developed and extensively used for analysing mean large-scale spreading of water masses or minor species from a known source over different time periods (e.g. Durgadoo et al. 2017; van Sebille et al. 2015; Rühs et al., 2013). In our study, the DBPs from ballast water discharge are approximated as particles that are passively advected with the

simulated flow. The streamline calculations are purely advective and no diffusivity is applied. For both regions of interest, ten individual simulations are conducted starting each year in January from 2001 to 2010. The ten different simulations are used to obtain more robust ensemble results and avoiding extremes from internal variability. In each simulation, particles are continuously released close to the port site at every model output time step (once every five days), which represents a continuous ballast water discharge at this location. Subsequently, the particle advection is simulated for two years. For the

purpose of calculating seasonal and annual means and to allow for an initial accumulation period, only the months 11 to 23 (December – November) is analysed for each simulation. Additionally, all particles older than eleven months are not considered in the analysis so that total particle number is constant at each time step.

The experiments were run for the Pearl River Delta (PRD) region and the Singapore region. The Pearl River Delta region

comprises three major ports, Hong Kong, Guangzhou and Shenzhen, for which we derive a total annual ballast water discharge volume of 271 million m³ (8 % of the global ballast water discharge) from the modified world port ranking (Supplement Table). The Singapore region comprises the ports of Singapore and Tanjung Pelepas with a ballast water amount of 190 million m³ (5.6 %) each year. The discharge location where particles are released has been chosen in the vicinity of the harbours at approximately 8 to 40 km off the coast, ensuring minimal influence of the coastal circulation on

the simulation. We assume that the DBPs are transported from the inner harbour into the adjacent coastal areas where our model simulations are initialised. In many ports this is reasonable since rivers and tidal flushing cause a steady turnover of coastal and estuarine waters with the ocean.

For the analysis of the experiments we distinguish 1) the passive spread of DBPs without any environmental sinks (hereafter

PASSIVE), and 2) the spread of bromoform as a major volatile DBP accounting for atmospheric fluxes and oceanic sinks (hereafter FLUX). For the PASSIVE analysis, we consider the full history of simulated particle positions which is equivalent to assuming no particles getting lost through sinks in the ocean or emission into the atmosphere. The resulting distribution shows where DBPs in ballast water or assumingly dimensionless and immotile species can be transported through ocean currents within one year.




For the FLUX analysis each particle is given an initial mass of bromoform based on the ballast water volume of the harbour and the produced bromoform according to the three scenarios, MODERATE, HIGH and LOW (Table 2). Moreover, different sinks of bromoform such as constant exchange at the air-sea interface or chemical loss rates are taken into account.

We calculate the bromoform air-sea exchange based on the flux parameterisation from Nightingale et al. (2000) for all particles that reach the mixed layer at a certain time step. The mixed layer depth (MLD) is defined as the ocean layer where

the vertical density gradient does not exceed 0.02 kg m$^{-3}$ referenced to the 10 m depth. According to ORCA, the annual mean MLD is less than 20 m deep within our research area. Since the results are given at a five-day temporal resolution and the MLD is relatively shallow, it is reasonable to assume that the whole mixed layer is in contact with the atmosphere at least once during each Lagrangian time step. Treated ballast water provides an additional source of bromoform to the environment adding to the natural bromoform occurring in the ocean and atmosphere. Given the additive nature of the ocean and

atmospheric terms in the air-sea flux parameterisation, it is possible to calculate the flux of the anthropogenic and natural bromoform portions separately. For our simulations, we only consider bromoform from ballast water treatment and apply the air-sea flux parameterisation to the anthropogenic bromoform in water and air. We have conducted sensitivity tests with an atmospheric transport model which shows that the outgassed anthropogenic bromoform is quickly advected from the sea surface to other areas and different heights. Therefore we can assume that anthropogenic bromoform in the atmosphere is

always zero at the ocean surface in the region of interest. The air-sea exchange is linear proportional to the gas transfer velocity of bromoform which depends on surface wind velocities and sea surface temperature and salinity. Surface wind velocities are taken from the NEMO-ORCA forcing data set DFS5.2 (Dussin, 2016).

Oceanic sinks are also taken into account, although negligible on the time scales considered in this study. These include degradation through halide substitution and hydrolysis with a half-life of 4.37 years (Hense and Quack, 2009), and

remineralisation with a half-life of 5.72 years (Hense and Quack, 2009).

The particle density distribution is calculated on a 1°x1° horizontal grid over the upper 20 m of the ocean (further mentioned as "surface"). The distribution is given as percentage per grid box of total particle number (PASSIVE) and as bromoform concentration in pmol L$^{-1}$ (FLUX). Statistical values are calculated over three grid boxes with the highest concentration

around the discharge location. Analyses on seasonal to interannual time scales were conducted by averaging and concatenating the simulations from the model years 2001 to 2010. For calculation of time series, we use the smoothed two-week (15 day) running mean of the concentration and emission rates from the three grid boxes around the discharge location for the three scenarios MODERATE, LOW and HIGH. Wind speed values from these boxes are also smoothed with a 15-day running mean in order to better show the seasonal to annual variations.



## 3 Surface spread of DBPs – PASSIVE

Figure 2 shows the relative particle density distribution of DBPs averaged over ten years released from the Pearl River Delta. We estimate the contour lines of the percentage of DBPs (30%, 50%, 70% and 90%) that are characterised by the highest particle density. The distribution shows that 90 % of DBPs have spread past Japan and the Korean peninsula and with the Kuroshio into the North Pacific, as well as southwards into the South China Sea towards the Philippines within one year. On average, 30 % of the DBPs with the highest density will stay southward of the Pearl River Delta along the coast and are now distributed in the Gulf of Tonkin west of the island of Hainan. There, highest relative particle density distribution reaches locally up to 3 % with respect to total DBP discharge.

For the Singapore harbour region, the relative DBP distribution averaged over the years 2001-2010 is shown in Figure 3. As for the Pearl River Delta, most of the DBPs stay in the close vicinity of the coast lines with highest relative density distribution of 4 %. On an annual mean basis, the 30 % of DBPs that are characterised by the highest particle density have been transported north-west wards and accumulate in the Strait of Malacca being in close contact with the coast lines of the Malay Peninsula and the island of Sumatra. DBPs within the 50-70 % distribution expand mostly into the Indian Ocean towards Sri Lanka, but a small fraction is advected into the South China Sea between Borneo and Vietnam and even into the Java Sea. The main driver for the mean state of DBP transport from Singapore is the Indonesian Throughflow generally directed westward through the different passages of the Indonesian Archipelago (Gordon, 2001).

For Pearl River Delta and Singapore, the areas of the 90 % of DBPs with the highest particle density expand over 5.0 and 8.6 million km$^2$, respectively, illustrating the large possible spread of longer-lived DBPs in ballast water. The size of the area and dominant direction of expansion is subject to variability on different time scales.

We investigate the interannual variations in the spread of DBPs by analysing the area extent of the 30, 50, 70 and 90 % of particles with the highest density (Figure 4) for the time period 2001-2010. Largest variations are found for the annual mean distribution of the 90 % area which expands over 6.6-10.2 million km$^2$ for the Pearl River Delta region depending on surface velocity strength in the area. The extent of the 30 and 50 % regions varies less on interannual time scales. Our results show that half of the longer-lived organisms and chemicals in ballast water can be expected to be spread over a relatively constant area of 0.5-1 million km$^2$ around the harbour, while the other 50 % are transported into a much larger region (up to 10.2 million km$^2$) that fluctuates depending on interannual variations of ocean surface transport.

Since a lot of the volatile DBPs will be emitted into the atmosphere and other short-lived non-volatile DBPs degrade in the ocean on relatively short time scales of weeks to months, the seasonal timescales are also of interest when evaluating main pathways of DBP distributions. Depending on the season of discharge, the dominant atmospheric winds and oceanic currents can vary substantially in strength and direction in the region considered. We calculate seasonal anomalies of the particle



density distribution for the time period 2001-2010 by subtracting the annual mean climatology from the seasonal mean climatologies.

Seasonal anomalies of the main pathways of ballast water spread from the Pearl River Delta region show a clear reversal of main spread from boreal winter (DJF) to summer (JJA) (Figure 5). Surface currents in the South China Sea are wind-driven

and seasonally affected by the northwest Pacific monsoon (Shaw and Chao, 1994). During DJF, the main pathway is towards the southwest with accumulation of DBPs west of Hainan Island and positive anomalies up to 9 %. There is a clear separation of these positive anomalies south of Pearl River Delta and negative anomalies north of this region. Furthermore, the area of the 90 % DBP distribution is located in a narrower band towards the coast during DJF. This anomaly pattern reverses in JJA. More DBPs are transported northward while there is less advection to the south. However, the northeast

winter monsoon prevails much longer in the Pearl River Delta than the southwest summer monsoon, which explains why in the annual mean the largest part of DBPs is advected southward. During spring (MAM) and autumn (SON) anomalies are less pronounced. In MAM, the anomalies are mostly positive around the discharge location which means more DBP accumulation along the coast and slower transport than in the annual mean due to weaker currents. The opposite happens in SON with negative anomalies around the discharge location, indicating that fastest transport occurs during SON.

A similar seasonality in DBP spread can be seen from discharge in the Singapore region (Figure 6). Here, close to the equator, the monsoon winds seasonally reverse from north-westerly winds in JJA to south-easterly winds in DJF. As a result, more DBPs are transported towards the northwest through the Strait of Malacca into the Indian Ocean in DJF and Singapore ports show a negative anomaly. As expected from the reversed winds in JJA, less DBPs are advected towards the west and more towards the east so that the DBPs can reach the Pacific Ocean. During SON, strongest positive anomalies can be found

in the southern Strait of Malacca. Then winds are transitioning and become very weak thus DBPs cannot be transported quickly and accumulate near the discharge location of Singapore. Lowest anomalies are found in MAM with a slightly enhanced accumulation of DBPs north off Malaysia.

**4 Concentration and emission of bromoform – FLUX**

The oceanic distribution of bromoform from ballast water treatment and its emissions into the marine boundary layer are

estimated from the FLUX analysis based on the simulated velocity fields from 2006 and the corresponding Lagrangian experiments. As shown in chapter 3, the interannual transport variability is small and therefore a one year simulation is sufficient to derive the representative emission estimates. Bromoform as a volatile gas can be outgassed into the marine atmospheric boundary layer, as long as it stays at the ocean-atmosphere interface. In the FLUX analysis, we calculate the bromoform outgassing rate for all particles within the mixed layer at every time step. We also calculate bromoform surface

concentration in the upper 20 m and the sea-to-air flux averaged over one year for the three scenarios MODERATE, HIGH




and LOW. For comparison we also calculate the bromoform concentrations that would prevail without outgassing into the atmosphere from the PASSIVE analysis.

We find that surface concentrations from the FLUX analysis are largely reduced compared to PASSIVE. In the Pearl River Delta region, bromoform only remains in the box around the discharge location due to the new input of ballast water at every
time step (Figure 7). Thus, the majority of released bromoform is instantly outgassed into the atmosphere, resulting in a relatively constant concentration of 10 pmol $L^{-1}$ in the MODERATE scenario around the discharge location, ranging from 21 pmol $L^{-1}$ (in HIGH) to 3 pmol $L^{-1}$ (LOW) (Table 3).

Also bromoform concentrations from Singapore ballast water stay much more centred around the discharge location when compared to the PASSIVE analysis without outgassing (Figure 8). Small concentrations of 1 to 2 pmol $L^{-1}$ can still be found
in the Strait of Malacca. Average bromoform concentrations around Singapore add up to 11 pmol $L^{-1}$ in the MODERATE scenario, ranging from 23 pmol $L^{-1}$ (HIGH) to 3 pmol $L^{-1}$ (LOW). Measurements of bromoform in this region showed elevated surface concentrations of up to 130 pmol $L^{-1}$ (Fuhlbrügge et al., 2016) most likely due to the combination of strong natural and already existing anthropogenic coastal sources. The additional bromoform input to be expected from ballast water discharge at Singapore would thus lead to a 9 % (18.5 %; 2.4 %) increase.


Evaluation of the time series shows that wind velocities are enhanced in the Pearl River Delta region with a strong seasonal cycle (Figure 9). Such strong winds cause high exchange velocities which in turn lead to high emission rates. The bromoform emission rate stays constant at 1690 pmol $m^2$ $hr^{-1}$ because everything discharged into the ocean is instantly outgassed into the atmosphere. Therefore, both oceanic concentrations and emissions into the atmosphere stay relatively
constant throughout the time period, independent of the wind variations. The bromoform emissions in the Pearl River Delta region range between 440 and 3530 pmol $m^2$ $hr^{-1}$ for the three different scenarios (Table 3). This flux is much larger than in the Singapore region where the average range is 250 to 1940 pmol $m^2$ $hr^{-1}$. Around Singapore, concentrations and emissions underlie a strong seasonality driven by the wind speed. Two times a year in summer and winter, wind velocities increase and cause bromoform emissions to increase as well. At the same time oceanic bromoform concentrations around the discharge
location drop due to the increased emissions and faster oceanic transport (Figure 9). For weaker winds, the bromoform response is reversed with lower emissions and higher oceanic concentrations.

Adding up the air-sea flux rate of bromoform over one year, we derive the annual air-sea flux of bromine (Br) resulting from ballast water treatment. At Singapore, the total Br flux ranges from 80 to 640 kmol (7 to 55 t) Br $a^{-1}$ from LOW to HIGH, which corresponds to an outgassing of roughly 85 % of the original 8 to 63 t Br produced as bromoform in ballast water
(Table 3). The remaining 15 % are transported from the port site into the open ocean where they are either eventually outgassed into the atmosphere or transported into the deeper ocean. In the Pearl River Delta region, the flux ranges from 136



to 1070 kmol (12 to 92 t) Br a$^{-1}$ which corresponds to an outgassing of 100 % of the Br produced from ballast water treatment.

Given that 85 % of Singapore and 100 % of Pearl River Delta ballast water bromoform is directly outgassed into the atmosphere, we expect that on average 90 % of the anthropogenic bromoform is quickly outgassed after ballast water discharge in the Southeast Asia region. Based on this assumption, we estimate the anthropogenic outgassing rate of bromoform at each of the 38 harbours in the modified world port ranking in Southeast Asia according to its calculated ballast water volume for a MODERATE scenario. These emission rates are calculated on a 1°x1° horizontal grid box closest to the

harbour so that the values can be compared to emission maps reconstructed from observations after Ziska et al. (2013) (Figure 10). Note that anthropogenic emissions are always positive (from ocean to atmosphere) since they were calculated with zero concentration in the atmosphere. The climatological emissions can have negative (atmosphere to ocean) fluxes where the reconstructed atmosphere has higher concentrations than the sea surface. Thus adding anthropogenic emissions to the climatology can theoretically reduce emissions. However, the climatology from Ziska et al. (2013) shows that

bromoform emissions in coastal areas are generally characterised by high positive emissions with 500 to 1000 pmol m$^2$ h$^{-1}$ where macroalgae act as efficient bromoform producers (Quack and Wallace, 2003) (Figure 10, left panel). When we add the estimated anthropogenic bromoform emissions from ballast water to the climatological emissions, many of the grid boxes clearly show a strong increase in emission, sometimes more than doubling the emission rates of bromoform (Figure 10, right panel). This is especially visible at very big harbours such as Shanghai, Singapore and the Pearl River Delta region where the

new emission rates exceed 1000 pmol m$^2$ h$^{-1}$. These regions appear as local hot spots of anthropogenic bromoform emissions. Most of these areas are characterised by heavy industry and other anthropogenic activities, resulting in strong emissions of greenhouse gases like methane and ozone. The expected additional source of bromoform to the atmospheric environment can perturb the oxidising capacity and thus the atmospheric lifetime of greenhouse gases and other pollutants (Saiz-Lopez and von Glasow, 2012). The atmospheric chemistry around large ports is highly sensitive to additional

emissions of volatile DBPs from treated ballast water.

## 5 Discussion and conclusion

We investigate the new source of halogenated disinfection by-products to the ocean and atmosphere, from the release of chemically treated ballast water. Over the next years over 75,000 ships have to install a ballast water treatment system to prevent the continued spread of harmful invasive species (David and Gollasch, 2015). As a side effect, halogenated DBPs at

high concentrations will be produced in ballast water and released into coastal waters (Werschkun et al., 2012; Delacroix et al., 2013). In particular, bromoform shows concentrations in undiluted ballast water up to a million times higher than in the natural environment.



Our simulations of the DBP spread from the Singapore and Pearl River Delta harbours in the PASSIVE analysis show that within one year about half of the DBPs discharged with the ballast water spreads fast in the surface ocean, while the other

half accumulates close to coastal areas around the discharge location with a relative abundance of 3 to 4 % of DBPs per 1°x1° grid box. The currents determining the DBP spread in Southeast Asia are seasonally influenced by Monsoon winds. In Singapore, the main driver of DBP transport throughout the year is the westward Indonesian Throughflow and most of the DBPs spread into the Strait of Malacca and the Indian Ocean. For the Pearl River Delta region, the majority of DBPs is transported south-westward during the northeast monsoon period in boreal winter and north-eastward during the southwest

monsoon period in boreal summer. Thus non-volatile DBPs can either spread over large areas at the sea surface or accumulate in specific regions, such as the accumulation of the DBPs from the Pearl River Delta in the Gulf of Tonkin. While interannual variations of the DBP spread are relatively small, the seasonal cycle in transport patterns leads to enhanced coastal accumulations depending on region and time of year.

Based on our simulations in the FLUX analysis, we expect brominated VSLS concentrations and emissions to increase

locally in regions with high industrial activity. Anthropogenic bromoform can locally add up to 23 pmol L$^{-1}$ (0.006 µg L$^{-1}$) in the HIGH scenario around the port sites. Our simulations assume that DBPs are transported out of the harbour, therefore providing a lower boundary of the environmental concentrations. Other studies like David et al. (2018) that use a port-based model approach to calculate the predicted environmental concentration, estimate higher bromoform values (e.g. 0.3 µg L$^{-1}$) due to the smaller areas considered and the missing air-sea exchange. Once ballast water treatment has been established

globally, in-situ measurements will be necessary to confirm if existing model-based results provide realistic estimates.

Our simulations reveal that bromoform emissions to the atmosphere can exceed 1000 pmol m$^{-2}$ h$^{-1}$ for the MODERATE scenario. This is caused by moderate to high wind speeds above 10 m s$^{-1}$, which occur especially in the Pearl River Delta. Here the exchange speed is sufficiently high so that all anthropogenic bromoform within the mixed layer is instantly outgassed into the atmosphere. Anthropogenic bromoform from ballast water discharge does not accumulate in the ocean but

is rather an immediate additional input of Br to the atmosphere. This new source can locally double the climatological bromoform flux, calculated from Ziska et al. (2013). Especially around big harbours like Singapore, Shanghai or in the Pearl River Delta, the bromoform emissions to the atmosphere are substantially larger than natural fluxes and can perturb atmospheric chemistry. For the HIGH scenario, emissions of up to 3500 pmol m$^{-2}$ h$^{-1}$ can occur which is in the range of highest natural emissions found in global shelf waters, however not exceeding reported maximum values of

4450 pmol m$^{-2}$ h$^{-1}$ (Quack and Wallace, 2003).

Over the next decades, the impact of brominated VSLS on climate and ozone depletion will increase due to changes in atmospheric transport and chemistry (Hossaini et al., 2015; Tegtmeier et al., 2015; Fernandez et al., 2017). Increasing VSLS production from anthropogenic activities needs to be investigated and monitored in order to quantify its input to stratospheric bromine. On a global scale the bromine emissions from ballast water treatment reaches up to 13 Mmol Br a$^{-1}$ in the HIGH





scenario. Compared to current estimates of background bromoform emissions of 2 to 10 Gmol Br a$^{-1}$ (see Ziska et al., 2013 and references therein) the anthropogenic bromine input is rather small amounting to 0.1 %. Thus from the current estimates presented here, we do not expect an impact of anthropogenic VSLS from ballast water treatment on the stratospheric ozone layer.

Ballast water however, is not the only anthropogenic source of DBPs to the coastal oceans. DBPs are also produced through

the oxidation of drinking water, waste water, sea water in desalination plants or cooling water in power plants (e.g. Jenner et al., 1997; Werschkun et al., 2012). In contrast to drinking water where by-products are strictly regulated (Richardson et al., 2007), chemical treatment of seawater or brackish water containing high levels of inorganic bromine is not monitored regularly although it can lead to much higher levels of brominated DBPs. Thus it is of interest to investigate in future studies the combined effect of anthropogenic VSLS from all types of oxidative water treatment to the environment.

**Author contribution**

JM wrote the manuscript, performed the simulations and created the output. ST developed the research question and guided the research process. BQ helped in the formulation of the research question and analysed water samples. SG and MD provided water samples. AB, SR and JVD provided the model data, set up the ARIANE environment and helped with the simulation. All authors took part in the process of the manuscript preparation.


The authors declare that they have no conflict of interest.

**Acknowledgements**

We thank Stephanie Delacroix for providing ship-based samples of ballast water. The authors would also like to thank Yue Jia for simulations of the atmospheric bromoform mixing ratios. This study is carried out within the DFG Emmy Noether

group AVeSH (A New Threat to the Stratospheric Ozone Layer from Anthropogenic Very Short-lived Halocarbons). The model data used for this study were kindly provided through collaboration within the DRAKKAR framework by the National Oceanographic Centre, Southampton, UK. We especially thank A.C. Coward, A.L. New, and colleagues for making the data available.



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



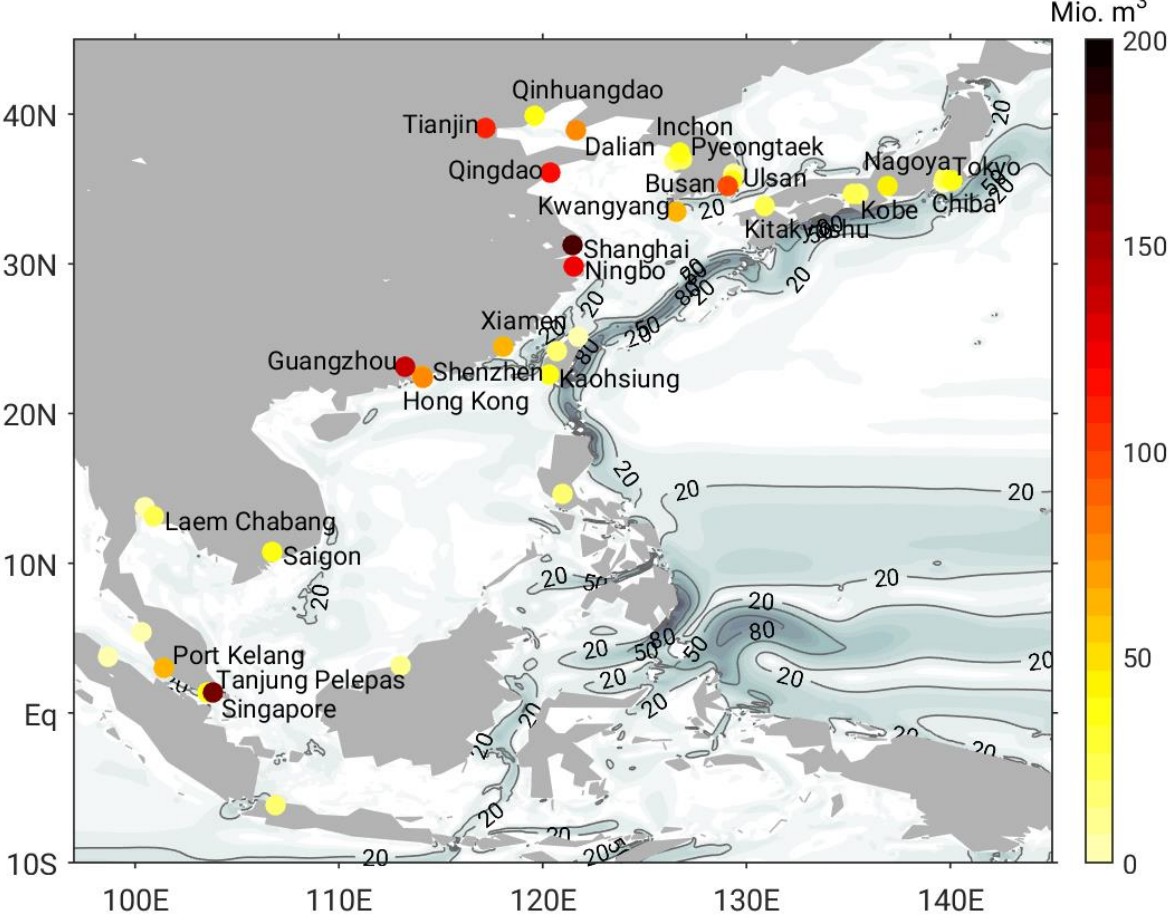

Figure 1: Estimated annual ballast water discharge volume [$10^6$ m$^3$] from each harbour in the modified world port ranking in Southeast Asia with the names of the largest 26 ports in the modified world port ranking (Supplement Table). Contours and black contour lines show climatological ocean surface velocities [cm s$^{-1}$] from NEMO-ORCA.



Table 1: Bromoform (CHBr$_3$) data from samples of undiluted ballast water given as average and standard deviation in nmol L$^{-1}$ and mg L$^{-1}$. Samples 1. and 2. are measurements from one chlorination BWTS, respectively. 3. and 4. are data from the IMO Marine Environmental Protection Committee (MEPC) reports on approval for different BWTS using chemical treatment.

| | CHBr$_3$ concentration [μg L$^{-1}$] | CHBr$_3$ concentration [nmol L$^{-1}$] |
|---|---|---|
| 1. Samples BWTS Norway | 244.5 ± 163.6 | 967.6 ± 647.4 |
| 2. Samples BWTS Germany | 202.0 ± 74.0 | 799.1 ± 292.7 |
| 3. MEPC report seawater | 239.4 ± 173.3 | 947.1 ± 685.8 |
| 4. MEPC report brackish water | 217.4 ± 155.1 | 860.0 ± 613.5 |
| **Mean** | **225.8 ± 141.5** | **893.5 ± 559.9** |




Table 2: Scenarios for simulation of ballast water (BW) spread with different initial bromoform concentration and annual bromoform amount for two regions in Southeast Asia, Singapore and Pearl River Delta (PRD)

| Scenario | $CHBr_3$ concentration in BW [µg $L^{-1}$] | Percentage of vessels using chemical BW treatment [%] | $CHBr_3$ Singapore [$10^6$ g $a^{-1}$] | [kmol $a^{-1}$] | $CHBr_3$ PRD [$10^6$ g $a^{-1}$] | [kmol $a^{-1}$] |
|---|---|---|---|---|---|---|
| MODERATE | 226 | 70 | 30 | 119 | 43 | 170 |
| LOW | 84 | 50 | 8 | 32 | 11 | 45 |
| HIGH | 368 | 90 | 63 | 250 | 90 | 356 |

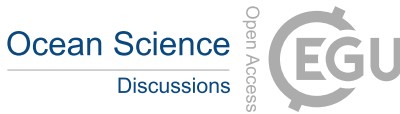



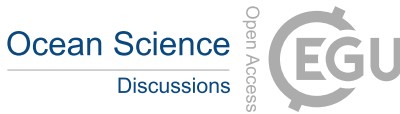

Figure 2: Annual mean surface (20 m) spread of DBPs from discharge at Pearl River Delta relative to the total number of particles released. Contours show the area of the percentage of particles (30%, 50%, 70% and 90%) characterised by the highest density.



Figure 3: Annual mean surface (20 m) spread of DBPs from discharge at Singapore relative to the total number of particles released. Contours show the area of the percentage of particles (30%, 50%, 70% and 90%) characterised by the highest density.





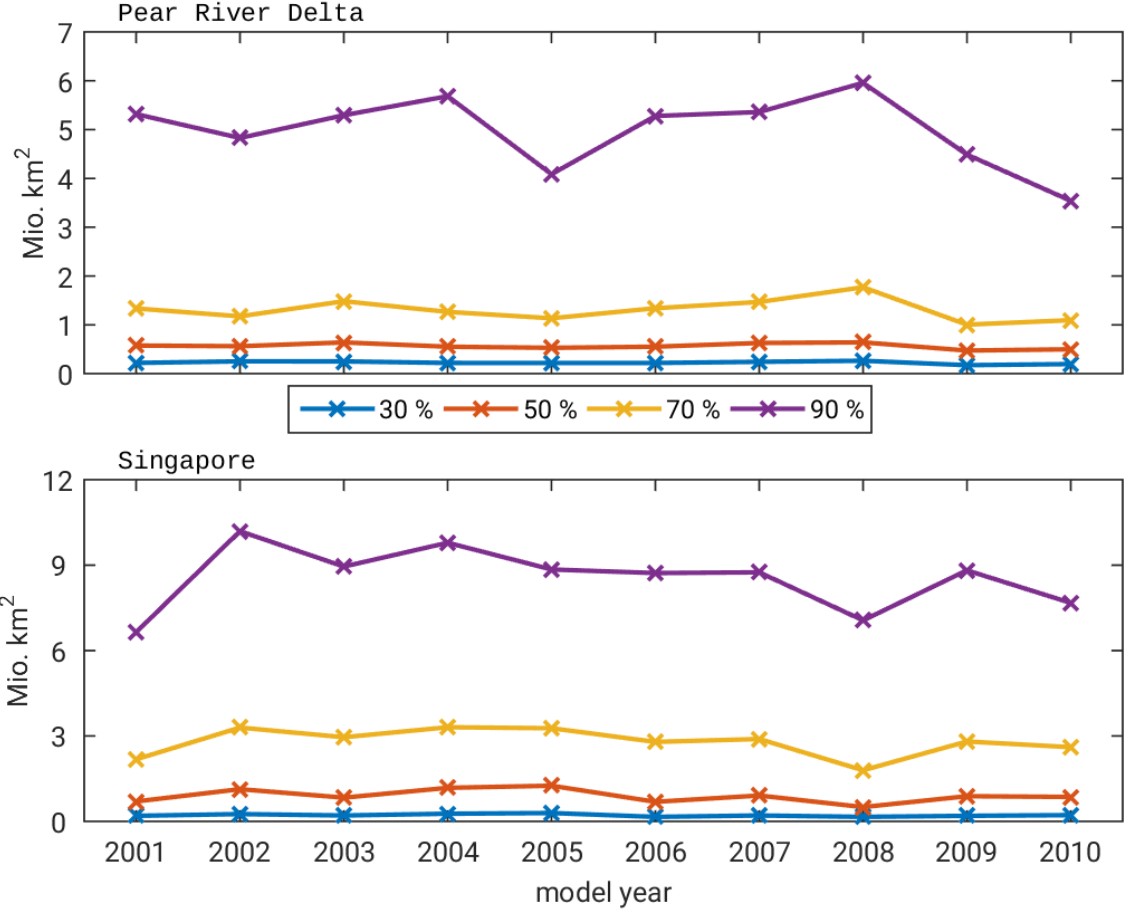

Figure 4: Annual mean area extent of DBP spread at (a) Pearl River Delta and (b) Singapore based on 30 %, 50 %, 70 %, and 90 % of the particles characterised by the highest density.



Figure 5: Anomaly of seasonal DBP spread compared to climatology (2000-2010) at surface (20 m) for discharge at Pearl River Delta. Black contour line shows the area of the 90% of particles characterised by the highest density in the seasonal mean.





Figure 6: Anomaly of seasonal DBP spread compared to climatology (2000-2010) at surface (20 m) for discharge at Singapore. Black contour line shows the area of the 90% of particles characterized by the highest density in the seasonal mean.




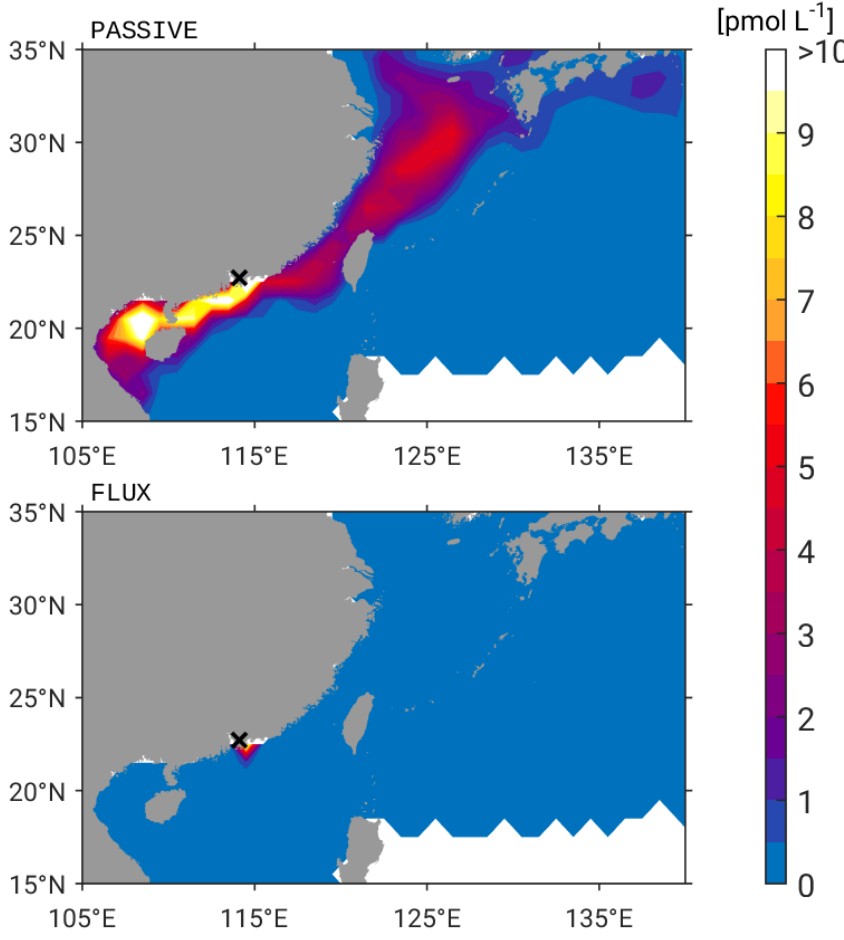

Figure 7: Surface bromoform concentration at Pearl River Delta for the MODERATE scenario, averaged over one year. (a) PASSIVE analysis without loss rates. (b) FLUX analysis with outgassing.





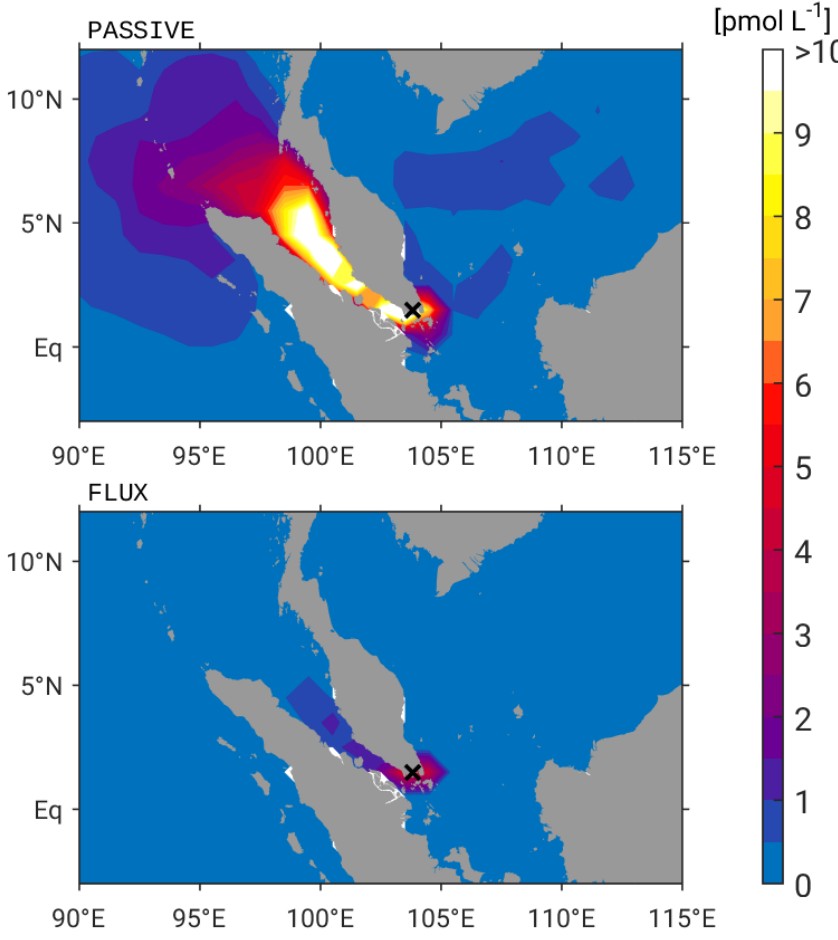

Figure 8: Surface bromoform concentration at Singapore for the MODERATE scenario, averaged over one year. (a) PASSIVE analysis without loss rates. (b) FLUX analysis with outgassing.






Figure 9: Time series of two-week running mean of wind speed (black), bromoform surface concentration (orange) and emission (blue) for (a) Singapore and (b) Pearl River Delta. Solid lines show values from the MODERATE scenario. Shaded areas show the range between HIGH and LOW scenarios for both bromoform concentration and emission.






Table 3: Average values for the FLUX experiment at Singapore and Pearl River Delta regions for different scenarios. Values are calculated as the sum from three grid boxes around the discharge location.

| Scenario | Singapore | | | Pearl River Delta | | |
|---|---|---|---|---|---|---|
| | Concentration [pmol L$^{-1}$] | Emission [pmol m$^{-2}$ h$^{-1}$] | Total Br flux [kmol a$^{-1}$] | Concentration [pmol L$^{-1}$] | Emission [pmol m$^{-2}$ h$^{-1}$] | Total Br flux [kmol a$^{-1}$] |
| MODERATE | 11 | 928 | 306 | 10 | 1687 | 511 |
| LOW | 3 | 246 | 81 | 3 | 448 | 136 |
| HIGH | 23 | 1943 | 641 | 22 | 3532 | 1070 |





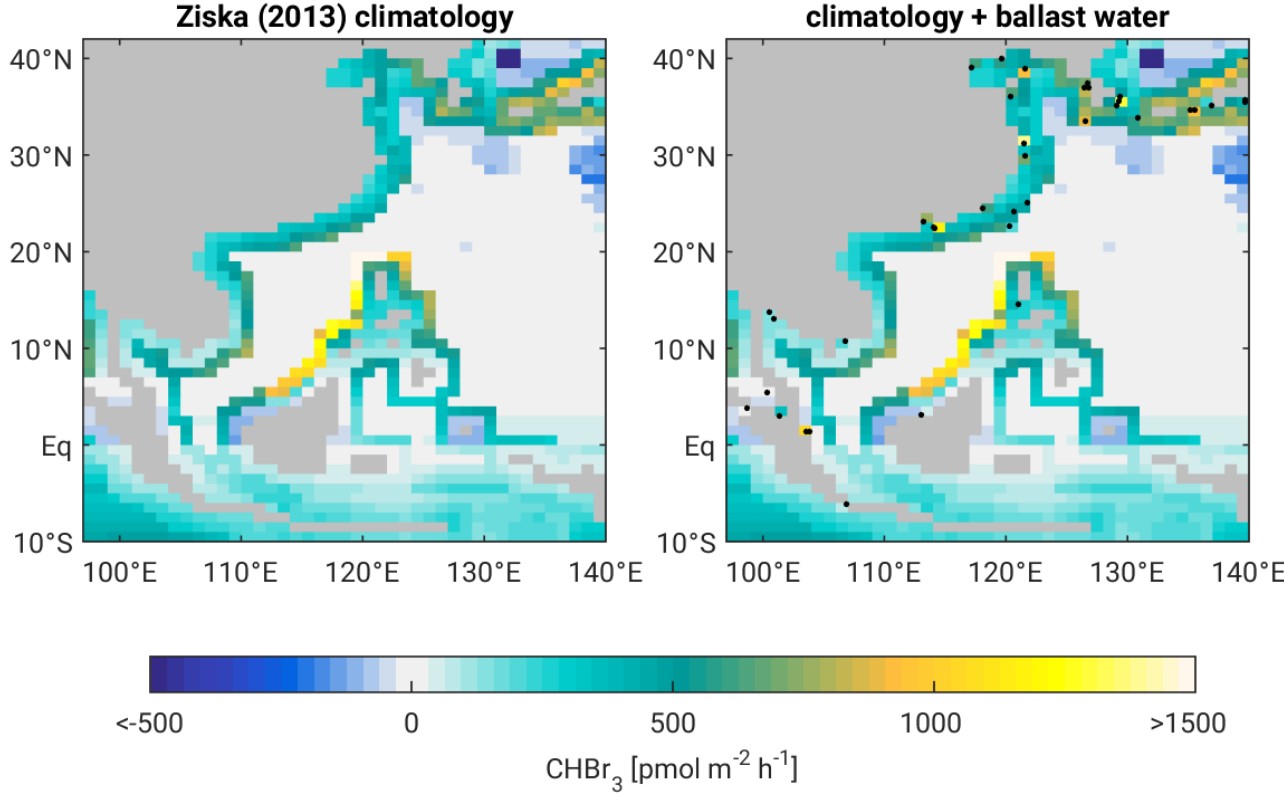

Figure 10: (a) Modelled bromoform emission rates updated from Ziska et al. (2013). (b) Same as (a) with additional anthropogenic emission rates calculated as 90 % of total bromoform release from ballast water treatment at each harbour. Black dots indicate location of all harbours from the modified world port ranking in Southeast Asia.