# Peer review of "Simulating the spread of disinfection by-products and anthropogenic bromoform emissions from ballast water discharge in Southeast Asia"

_Ocean Science, 2018_

## Referee Comment (RC1) · Anonymous Referee #1 · 11 Mar 2019

This manuscript extends the studies of environmental impacts of shipping by reporting the consequences of ballast water management convention to bromoform emissions and their dispersion in seawater. The study concentrates on South East Asia, but the methodology applied could potentially be applied also in larger scale. In my opinion, the paper is well written, and methodology is described enough if someone wants to repeat the work. I am not an expert of water pollution, which limits my analysis somewhat.

1. Why did you base you ballast water estimates on cargo statistics instead of using e.g. AIS data in your analysis? This would have allowed for an independent assessment

of BW discharge. This should be a fairly straightforward calculation if 1/3 of the cargo capacity is used to estimate the quantity of ballast water released.

2. Page 7, if BW is released during cargo operations, then assuming BW release 8-40 km off the coastline sounds a bit drastic. Liquid cargo operations may take place offshore, but I am not aware of container terminals this far out to the sea. If the motivation to put the emissions source out to the sea was "ensuring minimal influence of the coastal circulation on the simulation", why would this be the reason? I would imagine that if the harbour contribution to local levels of bromoform are a concern, then the harbour area would be a prime target for these investigations.

3. Page 13, lines 390-391: This new source can locally double the climatological bromoform flux, calculated from Ziska et al. (2013).

Somehow, I feel that this sentence sounds like a half of the story. I fully believe that authors are correct with their statement, but this should be put to context. It is easy to find locations where shipping is responsible for a very large portion of air pollution, close to 100%, if one chooses a location where ships are practically the only emission source. Still, it tells very little about overall situation. What I would like to see, is a discussion where bromoform emissions are put to a global context. This would help to see how big a problem DBPs are when compared to other climate contributions from ships.

OSD

---

## Referee Comment (RC2) · Anonymous Referee #2 · 8 Apr 2019

The paper addresses an additional source of bromine to the atmosphere from chlorination of ballast water. It is a first attempt to model the potential additional risk of ballast water treatment. The paper is well written and I recommend that the paper is published after addressing some issues. Up to date there are a number of ballast water treatments depending on chemical treatment (chlorination, chlorine dioxide, electrochemical treatment and ozonation) which are also mentioned by the authors. To extend the discussions regarding the magnitude of the emissions of brominated compounds from ballast water, the authors should consider to add discussions regarding

the usage of these different techniques, especially the chemistry of sea water ozonation. The composition of organic matter is also addressed by the authors. On a global scale – what are the authors take on the differences in bromoform production due to differences in dissolved organic matter? Also, the authors mention that environmental factors such as temperature and salinity might effect the formation of bromoform. It would be beneficial if the authors could expand this section to give a more complete picture of the magnitude of emissions on a global scale.

The global contribution of bromoform from ballast water is given to be 13 Mmol Br / year. How was this number derived? The total amount of ballast water on a global scale has for instance been estimated to be 7 billion tonnes on an annual basis. How would such an estimate influence the given annual flux?

---

## Author Comment (AC1) · 20 May 2019

Dear editor, dear referees,

please find attached the author comment on the reviews. The document consists of the authors' response and the revised manuscript including all documented changes.

Kind regards, Josefine Maas

Please also note the supplement to this comment:

[Figure]

https://www.ocean-sci-discuss.net/os-2018-151/os-2018-151-AC1-supplement.pdf

---

## Author Response (AR2)

**Author Comment on manuscript "Simulating the spread of disinfection by-products and anthropogenic bromoform emissions from ballast water discharge in Southeast Asia"**

Josefine Maas[1], Susann Tegtmeier[1], Birgit Quack[1], Arne Biastoch[1,2], Jonathan V. Durgadoo[1], Siren Rühs[1,a], Stephan Gollasch[3], Matej David[4,5]

[1]GEOMAR Helmholtz Centre for Ocean Research Kiel, Kiel, Germany
[2]Christian-Albrechts-Universität zu Kiel, Kiel, Germany
[3]GoConsult, Hamburg, Germany
[4]Dr. Matej David Consult, Izola, Slovenia
[5]Faculty of Maritime Studies, University of Rijeka, Croatia
[a]now at: Ocean Frontier Institute, Dalhousie University, Halifax, Canada

*Correspondence to*: Josefine Maas (jmaas@geomar.de)

We thank the editor for the comments about some changes on the manuscript. Please find below our response (in blue) to the comments, as well as the according changes. At the end, we added the revised manuscript with the marked-up changes.

**Topic Editor Decision:**

Comments to the Author:
I consider that your revision have addressed the Reviewers' comments satisfactorily, and have just a few small details to be addressed before publication. Line numbers refer to the version with Track Changes included in the author response:

1: Line 48: The radicals are presumably reacting with bromide; please make this clear

We added this in the text.

Lines 49 ff: "Ozone treatment forms hydroxyl and oxyl radicals which react with bromide ions in seawater to hypobromous acid or hypobromite ion which act as the disinfecting agents (Werschkun et al., 2012)."

2: Line 68, please explain which methodology was reviewed by GESAMP

The paper by David et al. (2018) investigated the method with which BWTS are tested to receive approval according to the Ballast Water Management Convention.

Lines 64 ff: "To receive approval with the Ballast Water Management Convention, BWTS need to perform a risk assessment according to the methodology of the Joint Group of Experts on the Scientific Aspects of Marine Environmental Protection – Ballast Water Working Group (GESAMP-BWWG) (IMO, 2017). The methodology tries to identify if the DBPs found in ballast water have an ecotoxicological effect on marine life as well as human health. The risk assessment uses a worst-case scenario where the DBPs discharged into the harbour are modelled to calculate their predicted environmental concentration. A recent study by David et al. (2018) showed, that the GESAMP-BWWG methodology does not fully account for the potential environmental risks. To date, the GESAMP-BWWG methodology has not considered environmental impacts on atmospheric chemistry from volatile DBPs either."

3: Line 84, the link between reactive halogens and HOx/NOx is not clear: why should oxidation of DMS and Hg affect HOx and NOx?

We clarified the atmospheric chemical reactions by reformulating the respective sentence as follows:

[revised manuscript text omitted]